# LOOKAHEAD-LSTM OPTIMIZER: A META-LEARNING K-STEPS METHOD

## ABSTRACT

All of the parametric machine learning models need to be optimized. At present, there are various optimization algorithms base on gradient, and they all have one thing in common—hand-designed. The purpose of Meta-learning is learning to learn, this thought can be used to do optimization automatically, without algorithm selection and hyper-parameters tuning. Previous works have trained a model to optimize other models using gradient information, but there still have some issues. We want to further improve the previous works in generalization and data transferability. then we propose **Lookahead-LSTM** optimization algorithm for this. [1]

## 1 INTRODUCTION

More and more machine learning problems can be cast as an optimization problem. Gradient information is widely used in continuous numerical optimization of machine learning model especially deep neural network. Besides being able to work in convex problem, hand-craft algorithm like stochastic gradient descent with momentum (Sutskever et al. (2013)), Adagrad (Duchi et al. (2011)), RMSProp (Dauphin et al. (2015)), Adam (Kingma & Ba (2015)) also can well handle non-convex, high-dimension problems which always appear in deep neural network.

But in the era of deep learning, no general rule tells us which algorithm is better than the others. Researchers and engineers generally choose algorithms according to some empirical rules, or customized for specific problems. In such an environment, researchers need to repeatedly try different algorithms and combinations of hyper-parameters to find solutions for specific problems. For deep neural network, training becomes a time-consuming phase. Hence we want to find a way to optimize numerical problem automatically.

In this project, we will improve the previous work of algorithm implemented by Andrychowicz et al. (2016), whose model is based on recurrent neural network and use LSTM as it's cell. For this model, every parameter that should be optimized shares a common LSTM $m$ (which we call coordinatewise LSTM) but has it's own trajectory (as a hidden state), and the gradient of each parameter work as input to LSTM $m$ in order to generate the updates. We can formal this process by:

$$\theta_{t+1} = \theta_t + g_t, \qquad g_t, h_{t+1} = m_\phi\left(\frac{\partial f(\theta_t)}{\partial \theta_t}, h_t\right)$$

We call LSTM $m$ as optimizer, and we denote $\phi$ as the parameters of $m$, $h_t$ as the hidden state of parameter $\theta$ in time step $t$, $g_t$ is the update of parameter $\theta$ in time step $t$. Optimizer must be trained in a given network which we call meta-optimizer.

Architecture transferability means the optimizer can optimize the different network from meta-optimizer. Data transferability expresses that optimizer has the ability to optimize dataset which is different from training phase's. Although LSTM can learn to transfer to different architecture

---

[1] A large part of this work was done in 2020.

from one hidden layer to two hidden layer, or different number of hidden units (neurons), but there are still some problems which we will introduce in section 2.2 of it.

## 2 BACKGROUND

### 2.1 RELATED WORK

The concept of Meta-learning or Learning to learn (Brazdil et al. (2009)) has been put forward for a long time, it comes from psychology and now is used for machine learning. Because of the differentiable neural network, end-to-end Meta learning system can be implemented in the present days.

Machine learning is widely applied in more and more fields, not only the application model, but also the parameter tuning which needs rich experience and professional knowledge. AutoML (Hutter et al. (2019)) aims to automate the process of machine learning. First, it can lighten the burden of researchers. Then, it cut down the carpet strip of machine learning for abecedarians. Finally, it tries to construct an end-to-end learning system without human intervention.

In recent years, many methods are proposed to learn optimization automatically. Andrychowicz et al. (2016) advance a new model which is called Learning to learn by gradient descent by gradient descent (L2L). This method attempts to simulate SGD with momentum and adaptive learning rate algorithm like RMSProp or Adam using LSTM. Li & Malik (2016) and Li & Malik (2017) aim the same target with Andrychowicz et al. (2016), but they use reinforcement learning but not LSTM, and they adopt guided policy search to adjust the parameters of optimizer. Wichrowska et al. (2017) partial solve the problem of architecture transferability by using hierarchical RNN and a set of problem including several tasks, which makes the model and training complex. Chen et al. (2017) consider a different situation when the target function is non-differentiable or a black-box. Hence the problem becomes how to balance exploitation and exploration. They consider it as a Bayesian optimization task and use RNN to simulate Gaussian process with a loss function using Expected Improvement.

Some applications have implemented this technique as well, Ravi & Larochelle (2017) rethinks the structure of LSTM and consider it as a process of optimization base on gradient, whose hidden state represents the parameters of optimizer, using gate as a momentum and learning rate, and gradient as input. They show that this model can improve the performance of few-shot learning.

### 2.2 EXISTED PROBLEM

For the model L2L, Andrychowicz et al. (2016) train an optimizer using a one hidden layer MLP activated by sigmoid as optimizer, and they find the optimizer can generalize to two layer MLP or one layer with different hidden units MLP, which has outperformed hand-designed algorithms. However, the process would fail since the loss never falls while the activation function changes to ReLU. Andrychowicz et al. (2016) tests and verifies only several small network, the generalization of L2L needs to be further tested.

Data generalization and transferability are of equal importance. When we use hand-designed algorithm like SGD or Adam, data type won't be taken into account, because of the generalization of those algorithms. L2L hasn't given the result of using different datasets in training and testing, like training with MNIST and testing with CIFAR-10, or vice versa.

Andrychowicz et al. (2016) train and test two types of network: convolutional neural network (CNN) and Fully connective network. But as an optimization method, it should be generalized to all of the numerical optimization problems. More tasks which can be optimized by gradient method should be tested with L2L.

## 3 METHOD

### 3.1 PROBLEM ANALYSIS

In this section, we analyze the above problem and try to deduce some conclusion.

Consider a simple toy regression model of one hidden layer MLP, we can formalize this setting as:

$$\mathcal{L} = \frac{1}{2}\sum_t (y_{true}^t - y^t)^2, \qquad y^t = \sum_i w_i z_i^t, \qquad z_j^t = f(\sum_i \theta_{ij} x_i^t)$$

in those equations, $\mathcal{L}$ is denoted as MSE loss function, $f$ is the activation function, and $\boldsymbol{\theta}$ are the parameters in hidden layer.

Then we derive the gradient $\frac{\partial \mathcal{L}}{\partial \theta_{ij}}$:

$$\frac{\partial \mathcal{L}}{\partial \theta_{ij}} = \frac{\partial \mathcal{L}}{\partial y}\frac{\partial y}{\partial z_j}\frac{\partial z_j}{\partial f}\frac{\partial f}{\partial \theta_{ij}} = \sum_t \left(y_{true}^t - y^t\right) w_j \frac{\partial z_j}{\partial f} x_i^t$$

And we find if $f$ is sigmoid function, we can derive the gradient:

$$\frac{\partial z_j}{\partial f} = f(\sum_i \theta_{ij} x_i^t)(1 - f(\sum_i \theta_{ij} x_i^t))$$

Then if $f$ is ReLU function, we can obtain it use same way:

$$\frac{\partial z_j}{\partial f} = \begin{cases} 1, x \geq 0 \\ 0, x < 0 \end{cases}$$

Hence we consider that L2L model might overfit to the special activation function, and if we employ a new activation function in optimizer, the optimization will fail.

According to the data transferability, there is the same situation with architecture because L2L may be overfit to dataset in training phase.

We think L2L has overfitted to the optimizer and dataset in training phase, including network architecture, activation function, loss function and data distribution. First of all, we need to verify the above analysis through experiments, then we will focus on how to improve the generalization of L2L.

### 3.2 SOLUTION

To solve these problems, we propose two different type solutions:

- The first solution is training optimizer with a more elaborate optimizer which includes more different activation functions, dropout, batch normalization or other structures. In order to improve the generalization of optimizer, we need to choice a representative network, but limited by resources, we just use a three layer network whose activation function both contain sigmoid and ReLU as optimizer to confirm our solution.
- The second solution is drawing lessons from transfer learning. If we have a well-trained optimizer by previous optimizer, now we want to use it to optimize another type network that has a quite different structure. We don't need to discard the optimizer and retrain a new one, we just train it a few steps using the different network structure as optimizer. This method likes transfer learning and it always be used in data transfer but we employ it on both data and model transfer.(Yosinski et al., 2014)

Due to the lack of computing resources and limited time, we just use simple experiments to verify the effectiveness of these methods.

## 3.3 IMPROVEMENT

We know the input of LSTM optimizer is the gradient of parameters, hence the quality of gradient would have significant impact to optimizer, if we can choice "correct" gradient and input to LSTM, it will be beneficial to LSTM training. But many optimization problems, especially those related to deep learning, are ill-conditioned. For those problem, gradient isn't a right direction, in many of there problem, gradient descent would often cause oscillate.

(Zhang et al., 2019) propose a optimization method named Lookahead, which aims to improve the stability of learning process and lowers the variance. This optimization algorithm assume a fast weight $\phi$ and original parameters(slow weight) $\theta$. There are two-lever-loop in this method, in the begin of inner loop, we assign slow weight $\theta$ to fast weight $\phi$ use a fast and simple optimizer(can be a hand-crafted) to operate $\phi$ $k$ times. Then we use $\alpha(\phi_k - \theta)$ to update $\theta$, $\alpha$ is a discount rate(in simple convex problem we can figure optimal $\alpha$, but it's impossible in complicated problem, like deep learning). This method can significantly reduce the oscillation.

Inspired by this method, we improve the source algorithm, we use LSTM optimizer which can learn how to find a proper $\alpha$. In training stage, we use $k$-steps inner loop, then input direction $\phi_k - \theta$ to LSTM optimizer, using back propagation to update parameters in LSTM. Testing stage is similar to training stage, we give testing algorithm in Algorithm 1. For the inner optimization algorithm $A$, we can choice simple and fast algorithm like SGD or SGD with momentum. It can also be use LSTM in inner loop, but this will lead to very slow running speed, so we do not recommend.

---

**Algorithm 1** K-steps Lookahead-LSTM optimization

---

**Require:** Initial optimizer parameters $\theta_0$, loss function $L$, dataset $D$, number of step $k$, inner optimizer $A$, LSTM optimizer $g$, training epoch T
  **for** $t = 1$ to $T$ **do**
    initialize inner temporary parameters$\phi_{t,0} \leftarrow \theta_{t-1}$
    **for** $i = 1$ to $k$ **do**
      sample a minibatch of data $d$ from dataset $D$
      update inner temporary parameters$\phi_{t,i} \leftarrow \phi_{t,i-1} + A(L, \phi_{t,i-1}, d)$
    **end for**
    update parameters and hidden states $\theta_t = \theta_{t-1} + g(\bigtriangledown L(\phi_{t,k} - \theta_{t-1}))$
  **end for**

---

# 4 EXPERIMENTS AND ESTIMATION

## 4.1 EXPERIMENTS

In experiments, we focus on how to generalize L2L optimizer to different network architecture. The preparatory work is to reproduce the optimizer model and its problem. We train optimizer using a two layer fully connective neural network whose activation function in hidden layer is sigmoid with MNIST dataset. After training, we test it in three networks whose activation function are sigmoid, ReLU, Tanh respectively.

The loss of LSTM in training phase is showed in Fig 1. The loss of of LSTM in one step is the accumulative sum of the loss of optimizer in $T$ times, in this experiment we set $T = 20$. We reinitialize optimizer network every 10 LSTM steps(which we call unroll step) because optimizer will overfit to a specific parameters combination without that.

The testing result(We use trained LSTM optimizer to train other network) is showed in Fig 2. As we expected, the training is fail in network with ReLU activation function. But to our surprise, network with Tanh activation function achieve good performance. We guess the reason for this is Tanh has similar trends as sigmoid.

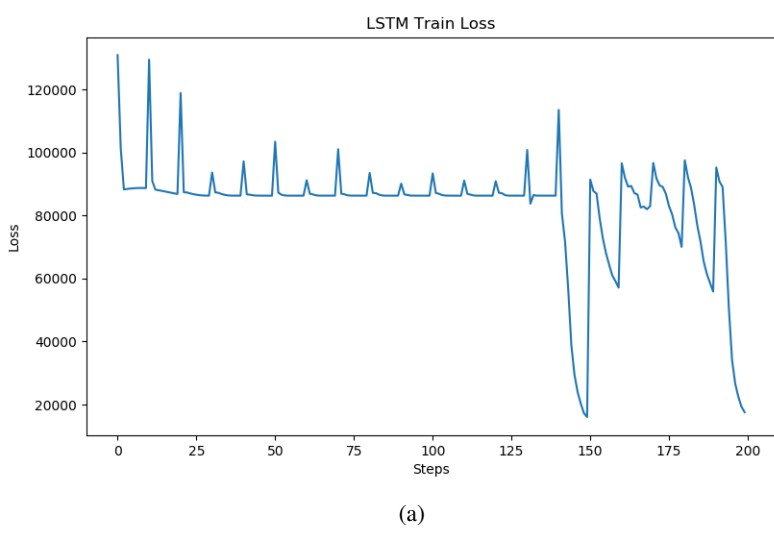

(a)

Figure 1: This figure is about the loss of training optimizer. The loss of of LSTM in one step is the accumulative sum of the loss of optimizer in 20 times. And in order to avoid overfitting the optimizer is unrolled in every 10 steps in the figure, so the pulse appear in every 10 steps. The optimizer is a two layer network with sigmoid activation function.

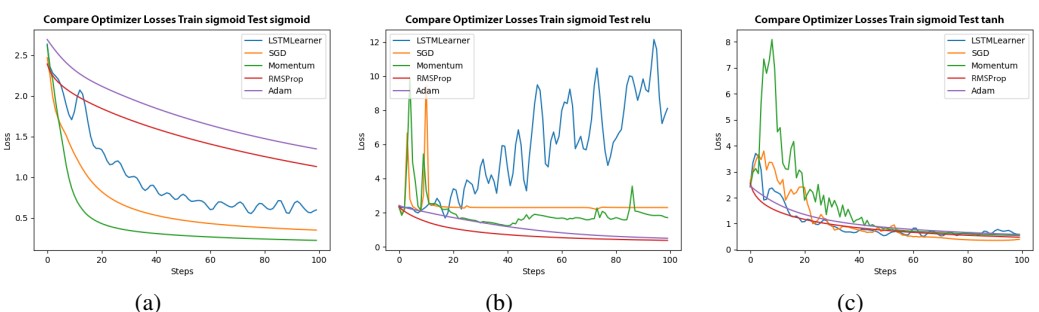

           (a)                        (b)                        (c)

Figure 2: These figures are optimizing different optimizer using various optimizer(Left: optimizer with sigmoid, Middle: optimizer with ReLU, Right: optimizer with Tanh), including SGD, SGD with momentum, RMSProp, Adam, and LSTM. We can find that LSTM fail in optimizer with ReLU activation function, and it don't perform well and occur oscillation in both network with sigmoid and Tanh activation function. We use these figures as our baseline.

In the first stage, we want to confirm the two methods above is effective. Hence we have two different experiments. One, we train optimizer again with a more elaborate optimizer. Two, we continue to train the optimizer trained in basic experiment with a different optimizer.

- The first experiment is about architecture overfitting, we replace the optimizer in training optimizer, the optimizer contains both sigmoid and ReLU activation function has three layers. We use those networks to train LSTM model and Independently test in the networks which sigmoid, ReLU, Tanh activation function respectively. Then we will compare to the performance in Andrychowicz et al. (2016) and estimate whether it's overfitting.

- The second experiment is about model transferability. Hand-designed optimization algorithm has the ability to generalize to different model. If we want to transfer the original model to a different network architecture, we can train the original LSTM optimizer using new optimizer for a few steps, like transfer learning, we shift the optimizer to target domain.

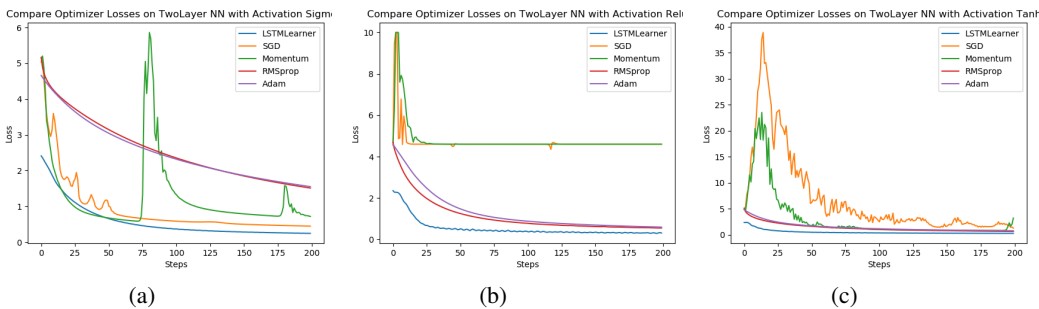

(a)                            (b)                            (c)

Figure 3: These figures are optimizing different optimizee using various optimizer(Left: optimizee with sigmoid, Middle: optimizee with ReLU, Right: optimizee with Tanh). The optimizer is trained by a three layer network whose activation function include sigmoid and ReLU, and optimizees are both two layer but have different activation functions. As we can see, the optimizer achieve best performance in all optimizee, even Tanh isn't appear in training phase. The LSTM optimizer obtain generalization in small network and different activation function.

In the second stage, we will attempt to improve the performance of L2L model using $k$-steps Lookahead. We can use this improvement to reduce oscillation in Figure 1. and increase convergence rate. We will compare with original LSTM and other hand-designed algorithms.

## 4.2 ESTIMATION

In estimation, training loss is our most concerned indicator, because it can obviously exhibit how fast the optimizer work and how low the final result is. In each experiment, we compare the results with those of hand-designed algorithms, including SGD, SGD with momentum, RMSProp, Adam. In basic experiments(Figure 2), we find that optimizer trained by network with sigmoid can't optimize network with ReLU. And it don't work well as SGD or SGD with momentum in network with sigmoid.

The most important evaluation method of an optimization method is to guarantee the convergence, but we can't give a formalized bound because the optimizer is a neural network. Hence we need to do a considerable number of comparative experiments.

For the training loss, we will judge the result in three aspects: rate of descend, Lowest point of convergence and others. High rate of descent means that optimizer needs to take less steps to bring the model to lower loss. Lowest point expresses that the capacity of optimizer and we can determine whether optimizer is easy to fall into local optimal solution. Others elements include oscillation, validation loss et al.

## 4.3 RESULT

We show result about two propose method in Figure 3 and Figure 4. As same as basic experiments, We compare the result with other hand-designed algorithms.

For the first method, we use a three network with both sigmoid and ReLU activation function as optimizer to train LSTM optimizer, and use it to train three two layer networks with different activation function. The result of this experiment is given in Figure 3 in three figures. In Figure 3(a) and Figure 3(c), The performance of LSTM is better than other optimizer and the original LSTM. For the ReLU activation function in Figure 3(b), LSTM optimizer finally converge and outperform than other optimizer. Hence we know this method can explain and overcome the problem of overfitting.

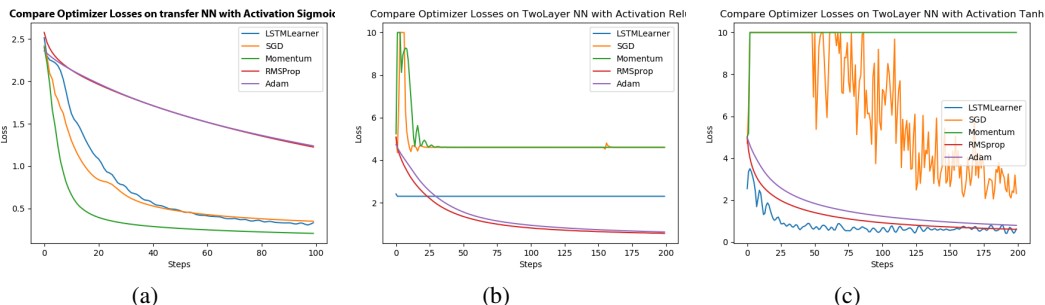

(a)                    (b)                    (c)

Figure 4: These figures are optimizing different optimizee using various optimizer(Left: optimizee with sigmoid, Middle: optimizee with ReLU, Right: optimizee with Tanh). LSTM optimizer fine tune by the network with ReLU, and we can find that it reduce the oscillation and marginally improve the performance of optimizer with sigmoid and Tanh. But it still fail in network with ReLU. In optimizee with ReLU, the LSTM optimizer is outperform SGD and SGD with momentum, but weaker than RMSProp and Adam

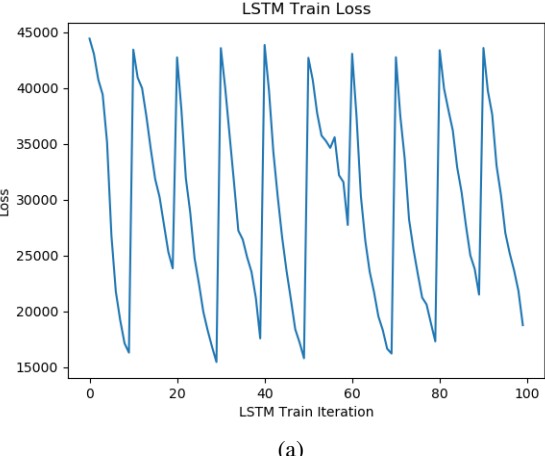

(a)

Figure 5: This figure is about the loss of training k-steps LSTM optimizer whose $k = 5$. In order to avoid overfitting the optimizee is unrolled in every 10 steps in the figure, so the pulse appear in every 10 steps. The optimizee is a three layer network with sigmoid activation function, and the numbers of layers are (400,200,10).

For the second method, we continue to train the original LSTM trained in basic experiment using new network, this network has different architecture, like activation function, task type(regression or classification), strategy(dropout), layer(batch normalization). In this experiment, we just select different activation function(from sigmoid to ReLU) to evaluate this method. We show the comparison in Figure 4. In Figure 4(a) and Figure 4(c), we find LSTM optimizer is close to SGD, but it's weaker than momentum. We conjecture the reason why transfer training fail, and we think the problem is in SGD, we review all the example of optimizing network with ReLU activation function(Figure 2(b), Figure 3(b), Figure 4(b)), we find that the worst case of optimizer is SGD, and LSTM optimizer base on SGD(it only use gradient as input), hence there is a high probability of failure in LSTM when SGD is failed.

In the second stage, we propose k-steps Lookahead-LSTM optimizer to reduce oscillation and variance and make it more stable. We use k-steps Lookahead to train a new LSTM model, which uses three layers network with sigmoid activation function as optimizee. Figure 5 reveal the training

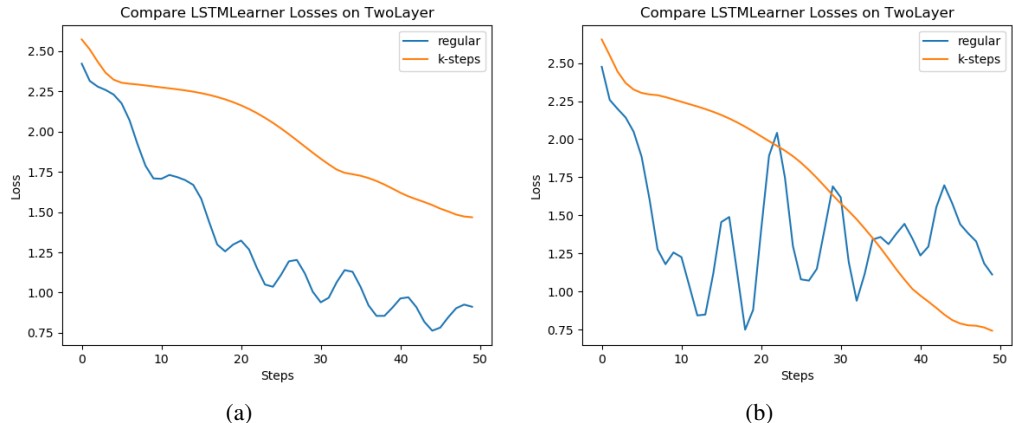

Figure 6: These figures are the comparison between original(regular) LSTM optimizer and k-steps Lookahead-LSTM optimizer. In Fig 6(a), we use two layers network with sigmoid activation function whose number of neuron is (20,10). In Fig 6(a), we use two layers network with sigmoid activation function whose number of neuron is (200,10). In both two figures, we can find that Lookahead-LSTM reduce oscillation in original LSTM.

loss of Lookahead-LSTM.

Then we compare the original LSTM(mentioned in Figure 1) and Lookahead LSTM in optimizing two layers network with sigmoid activation function. Figure 6 illustrate that Lookahead-LSTM can significantly reduce oscillation and variance.

And we also make a another comparison in these figures, we use a three layers network whose number of neuron is (20,20,10) as optimizer to train original LSTM, then use a three layers network whose number of neuron is (400,200,10) as optimizer to train k-steps Lookahead-LSTM. The optimizer in Figure 6(a) has two layers whose number of neuron is (20,10), and the other one has two layers and (200,10) neurons. As we can see, original LSTM is more outperform than Lookahead-LSTM in Figure 6(a) although it has some oscillation. But in Figure 6(b), we can observe that Lookahead-LSTM is better and more stable than original one. This phenomenon shows that LSTM optimizer is easy to overfit on the network structure(such as number of neuron in this case).

## 5 CONCLUSION

In machine learning, many problems can be changed to optimization problems, and then we can use hand-designed algorithm to figure out optimal solutions. But why can't we design a automatic optimization method.(Andrychowicz et al., 2016) propose a method which uses LSTM model to learn how to optimize neural networks.

In this project, we re-implement LSTM optimizer and try to generalize it's area of application. We raise two method to do it, one is using a more complicated network as optimizer, the other is transfer training. According to the result, we conclude that the first method is better than the second one.

In experiment, we find that LSTM optimizer would oscillate in training optimizer. Hence we employ a method names k-steps Lookahead to stabilize optimizer, we call it **Lookahead-LSTM**. We can observe that **Lookahead-LSTM** can reduce oscillation than original one, but it can't mitigate the symptom of overfitting. This will be an majorization direction in the future.

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
