# OpenReview forum: "Lookahead-LSTM Optimizer: A Meta-Learning K-steps Method"
_ICLR.cc/2026/Conference — ICLR 2026 Conference Withdrawn Submission_

### Official Review · Reviewer_DgnK · 2025-10-29

**Soundness:** 1
**Presentation:** 1
**Contribution:** 1
**Rating:** 2
**Confidence:** 5

**Summary:**

The submission aims to meta-learn optimizers parameterized by LSTM to eliminate the bias introduced by the hand-drafted optimizers. However, this is a very general motivation in the Learn to Optimize area. The submission failed to clarify their motivation and main contribution, with very limited novelty. Besides, the experiment settings are not diverse and concrete.

**Strengths:**

The osiliting problem in the meta-learned optimizers is interesting.

**Weaknesses:**

1. The writing is not appropriate for a lot of types, and a lot of sentences are confusing.
2. The motivation of the submission is not clear in terms of identifying the problems in the current learning to optimize the community.
3. The novelty is limited, and the references of the related work are old.
4. The experiment settings are not concrete; the stand L2O benchmarks are not utilized.

**Questions:**

The paper remains at an unfinished level. The method part is not well-written. What is the main technical contribution of this submission?
Is the proposed method able to defeat the handcrafted optimizer on the standard benchmarks, such as CIFAR-10 and CIFAR-100?
What is the runtime cost and training cost of the learned optimizer?
Can the learned optimizer from one domain be deployed to a novel domain?

---

### Official Review · Reviewer_3Ls2 · 2025-10-31

**Soundness:** 1
**Presentation:** 1
**Contribution:** 1
**Rating:** 2
**Confidence:** 5

**Summary:**

In this paper, the author identify some limitation in existing optimizers. Most optimizer are hand-designed, which is hard to adapt to different training scenario. In this paper, the author propose to use transfering learning techinique to learn an optimizer under different network. So the author use the LSTM network combine with k-step look ahead to form a new optimizer which can combine the experience of optimizer from other problems.

**Strengths:**

The overall idea is interesting. Designing a learned optimizer with transfer learning capabilities represents a novel and promising direction.

**Weaknesses:**

1: The current version of the paper appears to be in an early draft stage. It contains a significant amount of exploratory discussion and incomplete experimental results, and does not yet present a clear or solid conclusion.

2: Based on the description and algorithmic details, it is unclear whether this work truly introduces a meta-learning-based optimizer. The method appears to closely resemble the Lookahead optimizer, and I do not see clear elements of meta-learning or transfer learning as claimed. The authors should clarify what differentiates their approach from existing optimizers and how meta-learning is incorporated.

3: The experimental evaluation is not sufficiently convincing. In the reported results, a simple momentum-based optimizer performs better than the proposed method while also being more computationally efficient. More rigorous comparisons and stronger empirical evidence are needed to support the claims.

**Questions:**

Based on the current version of the paper, I would like the authors to clarify their method and clearly explain how it relates to meta-learning. In particular, please specify which component reflects a meta-learning mechanism and how it differs from existing optimizers such as Lookahead.

---

### Official Review · Reviewer_L13N · 2025-11-04

**Soundness:** 1
**Presentation:** 1
**Contribution:** 1
**Rating:** 0
**Confidence:** 4

**Summary:**

This paper addresses the issue of insufficient generalization of optimizers in learning to optimize, proposing meta-learning and transfer learning strategies across multiple training tasks, as well as introducing an improved K-steps Lookahead-LSTM method. Overall, the contribution is limited.

**Strengths:**

The authors attempt to improve the optimizer's performance on new tasks by increasing the diversity of training tasks and utilizing transfer training.

**Weaknesses:**

The proposed approach lacks sufficient novelty, and the experimental validation is not comprehensive. The writing of the paper also requires further improvement, as some sections are not clearly presented. Although it is claimed that this work was done in 2020, the references and compared methods are primarily focused on studies prior to 2020, without considering recent developments in this field after 2020.

**Questions:**

please refer to weakness.

---

### Official Review · Reviewer_WgZ6 · 2025-11-08

**Soundness:** 2
**Presentation:** 1
**Contribution:** 1
**Rating:** 2
**Confidence:** 4

**Summary:**

The paper proposes k-step LSTM optimizer, an effective lookahead optimizer for deep learning. They show performance improvements over alternative optimizers on a suite of tasks.

**Strengths:**

I think the paper is interesting in that it proposes a lookahead optimizer for deep learning tasks and reaffirms some of the motivations behind the motive of the original Zhang et al 2019 paper.

**Weaknesses:**

I feel the DL field is in general quite wary of yet another optimizer especially one whose ideas that have been more or less discussed in Zhang et al 2019, the paper lacks clarity in writing and requires more technical solidity in comparison.

**Questions:**

=== *clarity on LSTM* ===

I would love to clarity with the author that the LSTM refers to optimizing LSTM networks, not that the optimizer itself is making use of LSTM architecture? The presentation on this part seems a bit confusing in the early sections of the paper.

=== *baseline comparison* ===

Once again I think DL field is quite wary of a new optimizer so ablations are required to be highly rigorous. For all runs in the comparison eg Fig 2 there appears to be a single seed to the experiment and though the setups are generally about test time generalization, they also seem to be rather synthetic and would be of interest to test optimizers in more realistic scenarios (train on sigmoid test on relu does not sound very common in practice despite being a decent synthetic task).

All optimizers must be optimized heavily wrt their own set of hyperparameters and a clear explosion or pathological evolution of the loss seem likely to be a result of ill-tuned hyperparameters. In other words, the baseline must be tuned more heavily to make it a good comparison. There also does not seem to be comparison against Zhang et al 2019, and if I understand right the paper adopts quite similar idea so is just another application of the method? Can the author please clarify.

=== *losses* ====

As stated, in some comparisons, loss spikes seem related to ill-tuned optimizer hypers and it seems that LSTMOptimizer has lower starting loss  (fig 3, 4) compared to alternatives, any reason for that?

=== *presentations* ====

I think captions in various places Fig 6 can be improved to make the plots more informative. Also I think there are very simple typos in various places that need fixing.

---

### Note · Authors · 2025-11-13

I have read and agree with the venue's withdrawal policy on behalf of myself and my co-authors.